# The Homuncular Jigsaw: Investigations of Phantom Limb and Body Awareness Following Brachial Plexus Block or Avulsion

**DOI:** 10.3390/jcm8020182

**Published:** 2019-02-03

**Authors:** Mariella Pazzaglia, Erik Leemhuis, Anna Maria Giannini, Patrick Haggard

**Affiliations:** 1Department of Psychology, University of Rome “La Sapienza”, Via dei Marsi 78, 00185 Rome, Italy; erik.leemhuis@uniroma1.it (E.L.); annamaria.giannini@uniroma1.it (A.M.G.); 2Istituto di Ricovero e Cura a Carattere Scientifico Fondazione Santa Lucia, Via Ardeatina 306, 00179 Rome, Italy; 3Institute of Cognitive Neuroscience, University College London, London WC1N 3AR, UK; haggardpatrick@gmail.com

**Keywords:** phantom limb, somatotopy, brachial plexus, deafferentation rehabilitation, anesthetic block

## Abstract

Many neuropsychological theories agree that the brain maintains a relatively persistent representation of one’s own body, as indicated by vivid “phantom” experiences. It remains unclear how the loss of sensory and motor information contributes to the presence of this representation. Here, we focus on new empirical and theoretical evidence of phantom sensations following damage to or an anesthetic block of the brachial plexus. We suggest a crucial role of this structure in understanding the interaction between peripheral and central mechanisms in health and in pathology. Studies of brachial plexus function have shed new light on how neuroplasticity enables “somatotopic interferences”, including pain and body awareness. Understanding the relations among clinical disorders, their neural substrate, and behavioral outcomes may enhance methods of sensory rehabilitation for phantom limbs.

## 1. Introduction

Deafferentation, or limb loss, has a dramatic impact on the neural representation of the body [1], silencing all sensory and proprioceptive signals that flow in the primary somatosensory and primary motor cortex to enable limb control [2,3]. Deafferentation events often trigger sensorimotor changes in the brain; however, despite such reorganization, the missing limb remains strongly represented. Indeed, experiencing the sensation of a missing body part, which is known as a “phantom limb sensation”, is an extremely common condition in people experiencing any type of deafferentation—i.e., amputation, spinal cord injury, and other neurologic conditions [4]. These types of subjective experiences can sometimes be painful [5]. Symptoms typically emerge within days after an injury and can last for weeks, years, or even for the patient’s entire life [6]. In the best-case scenario, affected people can easily cope with immobilized [7], spatially misplaced, or telescoping limbs [8]. However, their quality of life and psychological health are often deeply compromised. A “side effect” of the challenge in providing relief and rehabilitation protocols for such patients is that clinical treatment effects also improve the scientific understanding of the mind-body (or body-mind) connection.

In some cases, such as those involving upper limb deafferentation, the “phantom” can be actively evoked through tactile stimulation of the ipsilateral cheek [4,9,10,11,12]. Phantom limb sensations elicited by the stimulation of body sites neurally adjacent to the affected limb are reported in 60–95% of individuals with amputated limbs [13,14]. Additionally, this phenomenon is frequently reported in cases of an intact limb with a loss of sensory and motor innervation, either due to nerve avulsion [15] or a spinal cord injury [16]. Thus, the occurrence of phantom limb sensations elicited by the stimulation of body sites adjacent to the affected limb appears to be caused by interrupted sensorimotor traffic, rather than the absence of a part of the physical body. Sensory referral of the affected (missing) body part onto a preserved body part is regarded as evidence of some persistent and stable representation of the deafferented body part [14], as well as evidence of somatotopically organized modifications [4]. This type of “somatotopic interference” has been widely studied, and it has produced new and advanced—but also controversial—insights into the cortical plasticity of body representations. The debate is related to the central versus peripheral origin of referred sensation leading to maladaptive plasticity taken over somatotopic map remodeling [5] or the altered inter-regional connectivity of the original, stable neural representation [17]. Interactions with neighboring somatosensory areas remain under discussion, and the sensory “homunculus” is not yet completely defined [18,19]. 

Phantom sensations have primarily been studied in amputees and, to a lesser extent, in patients with spinal cord lesions. However, such sensations can manifest in many other conditions and, interestingly, in a transient manner. Thus far, there is a paucity of research regarding the brachial plexus, which may instead serve as a model to investigate important questions regarding somatotopy, pain, embodiment, and neural reorganization. 

## 2. Phantom Sensation: A Perspective Following Brachial Plexus Block or Avulsion

The brachial plexus, which is a complex network receiving fibers from the cervical and thoracic nerves of the spinal cord, is completely responsible for the entire area of the upper limb [20]. Due to its topographical arrangement, following damage or anesthesia, it has the potential to permanently or temporarily interrupt the motor, sensory, and proprioceptive signals of the upper limb. The condition known as neonatal brachial plexus, which is linked to traumatic events during labor, is one of the more frequent causes of brachial plexus lesion, but little is known in relation to it about the presence of phantom sensations or about its impact on body representation in infancy [21,22]. In human adults, the role of the brachial plexus in the body began receiving attention in the 1960s due to the effects of blocking during surgical anesthesia [23]. The importance of this structure within the clinical field of neuropathic pain quickly became clear [24]. A pharmacological block of the plexus was then extended to treat acute pain caused by an avulsion injury or more distal damage, such as upper limb injury or amputation [25]. In healthy adults, a temporary anesthetic block of the brachial plexus can elicit an artificially induced phantom limb sensation [26]. A phantom arm sensation often fades when the effects of the anesthesia wear off [26]; however, in some cases, it can persist for a longer period [27]. 

An interesting aspect by which an artificial block or lesion of the brachial plexus may guide phantom phenomena is mislocalization. The classic work of Melzack and Bromage [26] showed that an anesthetic block of the upper limbs produces a significantly high rate of phantom arms. Interestingly, there is a different spatial orientation between the real limb and the phantom limb when sensation arises minutes after the injection. Furthermore, despite differences in the actual arm position between subjects, the phantom limb is often felt in a similar position. Specifically, during brachial plexus anesthesia, the phantom arm is experienced, in most cases, as resting on the chest/abdomen, even if it was on the operating table [26]. These observations were recently confirmed [27], and they suggest a highly different spatial localization between natural, pre-traumatic limb postural sensations, and post-trauma phantom sensation. The common experience of the phantom arm resting on the chest suggests common morphological and functional rearrangements after deafferentation, and it provides an important clue to the common bases of phantom phenomena.

Even more relevant are the somatotopic-type changes of phantom sensations observed in cases involving a brachial plexus injury, where patients commonly report a remapping of sensation from the hand to the cheek [4], in the jaw and buccal region [28], or in the ear [18]. Patients with total brachial plexus avulsions reported phantom limbs more frequently than patients whose lesion involved only portions of the brachial plexus [29]. Interestingly, these limb sensations were reported to be evoked by light touch on the surface of the external ear within days of the lesion [18], whereas they were reported on the jaw and buccal region after a number of months [28] and on the cheek area after a number of years [4]. This temporal pattern suggests that sensation from the deafferented limb may undergo a continuous and gradual process of neural reorganization onto a circumscribed region of the facial skin, with each spatial stage of remapping having a characteristic time scale. While many topographical relationships between somatotopical representations have been identified [30,31], the ear images of the sensory and motor homunculi, relative to other facial structures, remain under debate, leaving a hole in the homunculus. A limited number of studies found and confirmed, using electric stimulation of the ear on three different points, the presence of a specific area in the somatosensory cortex corresponding to the ear [32,33,34]. Using both magnetoencephalography [32] and functional magnetic resonance [33], the representation of the ear was located in the primary somatosensory cortex in both the neck and face areas. Much less is known about the motor cortical representation of the ear. Yu et al. [35] induced movement of an ear via electrical stimulation of the contralateral posterior portion of the superior temporal gyrus. The authors were highly cautious and suggested the presence of a connection with a “ear motor center” in the frontal areas in the stimulated structure, which has not yet been found in humans, but is located in primates [36]. However, clinical observations of ear remapping after a traumatic nerve injury suggest that such sensations can be specific to the upper limb [37], fully compatible with the hypothesis of a collocation of ear contiguous to the upper limb in somatosensory cortical maps. More recently, a single case study on a patient who experienced a brachial plexus avulsion injury was undertaken to better define and depict the hand in the ear [18]. After brachial avulsion, the patient reported that phantom limb sensations in the hand and arm were evoked by stimulating the ear. No effects from stimulation of the face were found. Tactile stimulation by light touch to the aural territory innervated from a branch of the vagus nerve revealed a high spatial specificity of phantom upper limb sensations. Stimulation of different points on a selected ear region evoked clearly delineated phantom digit sensations for either the dorsal or volar skin surfaces of the limb. Different forms of stroking delivered to the ear (e.g., continuous versus intermittent) were felt as the same sensation on the limb. Four weeks after the injury, even if the phantom ear sensation has vanished, the same systematic somatotopic changes can be experimentally re-induced using the rubber hand illusion framework [38]. This spatial specificity of referred sensation is fully compatible with the hypothesis of “somatotopic interferences” in the homuncular map of the somatosensory cortex. This possibility is strongly supported by the study of somatosensory-evoked potential [18]. In patients, stimulation within the vagally innervated territory of the external ear modulated activity within the centro-parietal regions, in accordance with the findings of previous studies [32,33,34]. The case demonstrates an important contribution to recent medical advances related to the vagus nerve, given its apparent role in bodily awareness [39] and its proven modulation of the centro-parietal cortical regions [40,41]. The phenomena, mechanisms, and neural structures involved in vagal stimulation have been explored in deafferented patients [42,43]. The auricular branch of the vagus nerve projects to the nucleus tractus solitarii. The direct pathways connecting the solitary tract nucleus and thalamus reach the parietal cortex and may initiate loop-like enhanced activity between the body and brain [42]. Although the role of the vagal network in bodily awareness remains unknown, vagus nerve stimulation seems to potentiate the spread of cortical signaling via the re-establishment of the thalamocortical network. Cortical responsiveness can, therefore, be modulated from short- and long-term modifications mediated by neurotransmitter release (e.g., norepinephrine and serotonin) and may evoke cortical reorganization, such as that which has been reported in the deafferented somatosensory areas of primates [44]. A recent work proposed a cortical multi-determined body model consisting of different representations of the cortical layers of the somatosensory cortex [45]. This interesting theoretical proposal attempted to answer the question, “Is the somatosensory cortex a sensory map or an animatable avatar?” Layer 4 of this primary cortical area can be considered a container of the body’s memories, resulting from genetics and experience. A puppet-like body model that goes beyond simply an up-to-date sensory map, it is a more stable and long-lasting model used to evaluate and project motor responses before final action production [45]. Especially in the somatosensory cortex, deafferentation is expected to imbalance this finely tuned interaction and stabilize body representations that cannot be correctly updated. Cortical representations likely do not change, as was previously thought. The stable topography, despite the presence of a brachial plexus avulsion [46], even decades after injury, emphasizes the need to determine what happens to the abolished somatosensory territory involving sensory and motor communication between the residual arm and the central nervous system [46]. Additionally, changes in functional brain organization after brachial-plexus injury could extend beyond the sensorimotor primary cortex [47]. Some features of the disorder, such as body-balance alterations, indicate the possibility of much larger neurological effects [48]. Moreover, it is possible to suppose that the effects of neurological trauma could have more connection than previously thought to many body-representation disorders that are typically considered to be of only psychiatric relevance [49]. In her work [50], Crawford supports the idea that, if the body schema can include missing body parts, it may also fail to recognize an attached and healthy limb as part of the body [51]. Altered parietal activity has been found in association with psychiatric disorders, such as xenomelia (the rejection of parts of one’s own body and the desperate desire for amputation) [52]. This supports the presented theoretical perspective and the need to extend physiological and neurological research to disorders typically understood as psychological [53].

For the moment, the mechanisms underlying the development of spontaneous or actively elicited phantom sensations remain hypothetical, as are those supporting the embodiment processes and the plasticity of bodily representations. In amputees, when a hand is deafferented, the neuronal territory dedicated to the hand in S1 can apparently be remapped with adjacent body representations [54,55]. In monkeys, after prolonged sensory deafferentation, the cortical representation of the body part is reorganized into neighboring areas [56]. Nevertheless, studies on humans did not yield similar results. The human brain does not exhibit the same topographical mapping as the primate brain [57,58]. Recent neural evidence documented the stability of somatosensory topographies and functional organization, despite massive sensory deafferentation [59]. Additionally, a phantom limb is occasionally reassigned to body parts that do not obey the rule of adjacency in cortical topography [60]. Corporeal shifts occur, for example, for the healthy contralateral hand, foot, or chest. The current phantom limb dogma also centrally undermines subcortical activity. Reorganization plays a large role in the potentiation and formation of new pathways in subcortical and peripheral structures [59]. The cuneate nucleus in the brainstem is likely a key point for reorganization following deafferentation [61]. Incoming facial projections, which from the trigeminal nucleus of the brainstem sprout and grow into the cuneate nucleus [62], could explain the presence of somatosensory interference between the referred sensations to the face and hand, but not as distinct from the ear.

While referred sensations from the ear/face to phantom limb experiences might well reflect peripheral and central processes, the comprehension of phantom pain could potentially disentangle the intricate picture. In many accounts, phantom sensations and pain both emerge from altered activity in the central nervous system caused by the loss of peripheral information in a top-down process of perceptual prediction. A different, coexistent, and less-explored approach is the “bottom-up” hypothesis [63]. This view attempts to answer the question, “How do peripheral nervous structures impact perception”? Amputation or nerve lesions activate regenerative processes. This activity often produces tumor-like structures, known as neuromas, in the stump area. The role of these structures in painful sensations has long been known and is well-established in scientific research; both spontaneous activation and active pressure on the stump are highly correlated with the onset of pain [64]. Despite the role of stump neuromas, both acute and chronic treatment of the phantom pain targeting the stump have had limited success. Moreover, specific clinical guidelines do not exist due to the high levels of individual variability. The limited efficacy of distal treatment has led researchers to focus instead on the central nervous system. Vaso et al. [63], however, supported the role of the peripheral nervous system with their study on dorsal root ganglia. An anesthetic block of the roots suppressed painful phantom sensations in many patients. The authors explained that this result reflected the suppression of ectopic neuronal activity. This view contrasts with top-down theories, which consider phantom pain a possible consequence of cortical reorganization induced by the lack of afferent information. Interestingly, non-painful phantom limb sensations were also affected by a peripheral block [63]. However, the possibility of evoking the phantom limb with tactile stimulation of cortically adjacent, but not peripherally adjacent, body parts and the high variability of non-painful sensations (i.e., telescoping, reference, and unnatural orientations), both suggest a greater involvement of cortical reorganization [63]. 

## 3. Conclusions

As has been briefly shown, many questions remain, and there may be more than one mechanism leading to phantom pain and the remapping of phantom sensations. A better understanding of phantom sensations in body representations will have a great impact on clinical and rehabilitative approaches to motor and somatosensory impairment. Although nerve grafting alone may have led to a resolution of both the phantom pain and the referred sensations, non-invasive interventions, such as mirror therapy [28,65], virtual reality [66], mental imagery [67], and other sensorimotor experiences, would be integrated to partially reset the equilibrium. Technologic advancements are proceeding quickly, and they are poised to offer many solutions regarding the maintenance of a functional or physical body. However, the necessary knowledge of both peripheral and central nervous system processes linked to deafferentation is relatively poor. Most advanced experiences with myoelectric and neural interfaces use muscular proprioceptive information [68] or artificial sensory feedback [69] to control active robotic prostheses. The results are encouraging and are already producing positive impacts in daily life [70]. A truly natural experience, however, remains a distant goal. A brachial plexus avulsion injury and brachial plexus anesthesia can represent effective models for studying deafferentation effects on the nervous system. Understanding adaptive brain plasticity and its impact on the overall perception process may guide the development of multimodal interventions focused on phantom limb control, as well as the exploitation of residual motor ability and sensory afferences [71,72]. Brachial plexus damage has a higher incidence rate, a more selective impact on health, and a greater promise of recovery, relative to other deafferentation conditions, such as amputations or spinal cord injuries. Therefore, studies of brachial plexus damage may be scientifically important to deal with the neural mechanisms of phantom limb sensations and pain, which, having been made in a single clinical case, obviously need to be confirmed to attain further knowledge. To conclude, we support an integrated view of the bottom-up and top-down contributions to the field of phantom sensations, body representations, and post-traumatic reorganization. A more detailed scientific understanding will require new experimental perspectives [63], including a wider range of clinical or artificially induced conditions [18,27] and a lucid theoretical reorganization of experimental results, as proposed by Brecht [45].

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
