# Peer review of "The Homuncular Jigsaw: Investigations of Phantom Limb and Body Awareness Following Brachial Plexus Block or Avulsion"

_jcm, 2019, doi:10.3390/jcm8020182_

Round 1
Reviewer 1 Report
I thoroughly enjoyed reading this paper. I still lean more towards the top down side of the scales but this paper is welcomed because there is clearly a balance required in the two camps. Clinicians could do worse things than contemplate the complexity of this avatar and the intricacies of its development and maintenance. I have nothing else to say except: Minor comment page 4 line 154 i dont think you can say that any neuronal territory in S1 "controls" the hand.
Author Response
Point 1: I thoroughly enjoyed reading this paper. I still lean more towards the top down side of the scales but this paper is welcomed because there is clearly a balance required in the two camps. Clinicians could do worse things than contemplate the complexity of this avatar and the intricacies of its development and maintenance.
Response 1: We are particularly happy this reviewer liked our work considering his/her careful reading of the MS and very well-argued comments.
Point 2 I have nothing else to say except: Minor comment page 4 line 154 i dont think you can say that any neuronal territory in S1 "controls" the hand.
Response 2: We thank the reviewer for this suggestion. This has been changed in the text.
The text on page 4 reads as follows:
"In amputees, when a hand is deafferented, the neuronal territory dedicated to hand in S1 can apparently be remapped with adjacent body representations"
Reviewer 2 Report
This article makes a contribution to ongoing research regarding the relationship between the peripheral and central nervous system, in part by exploring how the brachial plexus can affect phantom limb sensations. It utilizes multiple points of application including somatotopic mappings and processes, improved awareness of the importance of subcortical and peripheral structures in top-down versus bottom-up processing, cortical responsiveness, and adaptive brain plasticity. Overall, this article contributes to the theoretical re-conceptualizations of the workings of the somatosensory cortex and homuncular mappings when peripheral nervous system structures are also included.
In furthering their research on this topic the authors might consider looking at the literature on the neurodevelopment of the brachial plexus, and at studies of body image/body schema in terms of phantom limb sensations, and also cases of body integrity identity disorder (BIID) including apotemnophilia.
Author Response
We thank the reviewer for the careful reading of our paper and for the valuable suggestions.
Point 1: On the neurodevelopment of the brachial plexus, on page 2 we now say:
The condition known as neonatal brachial plexus, which is linked to traumatic events during labor, is one of the more frequent causes of brachial plexus lesion, but little is known in relation to it about the presence of phantom sensations or about its impact on body representation in infancy (Chang, Justice, Chung, & Yang, 2013; Socolovsky et al., 2016).
Point 2: On apotemnophilia, on page 4 we now say:
Additionally, changes in functional brain organization after brachial-plexus injury could extend beyond the sensorimotor primary cortex[47]. Some features of the disorder, such as body-balance alterations, indicate the possibility of much larger neurological effects[48]. Moreover, it is possible to suppose that the effects of neurological trauma could have more connection than previously thought to many body-representation disorders that are typically considered to be of only psychiatric relevance[49]. In her work [50], Crawford supports the idea that, if the body schema can include missing body parts, it may also fail to recognize an attached and healthy limb as part of the body[51]. Altered parietal activity has been found in association with psychiatric disorders such as xenomelia (the rejection of parts of one’s own body and the desperate desire for amputation)[52]. This supports the presented theoretical perspective and the need to extend physiological and neurological research to disorders typically understood as psychological[53].
Point 3: Finally, we admit that we encounter difficulty in assigning phantom-limb sensations to a disorder of the body schema or of the body image. This challenge is complicated by the inconsistencies, controversies, and variety of understandings regarding the significance of the concepts body image and body schema as diversely interpreted, sometimes even with opposing meanings, by various authors (Haggard and Wolpert 2005; Gallagher 2005; Giummarra et al. 2008; Mayer et al. 2008). It has recently been proposed that the brain integrates offline and online body representations, which sidesteps the dichotomy and contraposition of body schema/body image(Carruthers, 2008). The concept of online and offline body representations distinguishes between the body as it is constructed moment by moment and the body as conceived by relatively stable representations. The offline representation of the body may be useful for explaining phantom phenomena as a failure to record the affected limb in the offline body representation. Furthermore, a failure to record one body part in the innate offline representation might account for the existence of the body-part rejection phenomenon in xenomelia, in which one desires the removal of a limb that is obviously present in the online representation of the body. However, the evidence is still lacking for distinct neural mechanisms for representing both the online and offline body.